# Temporal Parallelization for GPU Acceleration of Spiking Neural Networks

## Abstract

Inspired by neurobiological structures, Spiking Neural Networks (SNNs) are heralded as a significant advancement in deep learning, given their potential for superior computational efficiency. However, this potential often remains untapped on contemporary hardware platforms. Specifically, when deployed on standard GPUs, SNNs tend to require extended computation times, placing them at a disadvantage compared to traditional Artificial Neural Networks (ANNs). Such inefficiencies have somehow diminished enthusiasm for SNN research and presented the tangible challenge to achieving scalability. To address such a challenge, this study introduces a temporal parallelization method specifically tailored for accelerating the propagation dynamics of SNNs on GPUs. Furthermore, we furnish two distinct implementations[1] based on the CUDA and JAX frameworks respectively, ensuring adaptability across both single and multi-GPU setups. When benchmarked against several established SNN implementations, the empirical analysis confirmed the efficacy of our proposed method. Notably, with the Leaky Integrate-and-Fire model as a test case, the CUDA-based implementation achieved $5\times$ to $40\times$ acceleration on the A100 GPU.

## 1 Introduction

Deep learning has firmly entrenched itself as a transformative field in computer science and artificial intelligence. Python, as a preferred programming language, has propelled this transformation. Notably, PyTorch (Paszke et al., 2019) stands out in the Python ecosystem, offering a robust platform for researchers working on traditional artificial neural network structures. Building on this momentum, the deep learning framework Flax (Heek et al., 2023), which leverages the capabilities of JAX (Bradbury et al., 2018), was introduced, encapsulating the high-performance programming features of JAX and further elevating the efficiency of deep learning development.

As the deep learning landscape evolves, the integration of biological neural dynamics, manifested in the form of Spiking Neural Networks (SNNs), has emerged as a promising research direction. Presently, the spiking attribute is primarily viewed as a module or characteristic that can be integrated into existing artificial neural networks (ANNs). This perspective is largely influenced by studies focused on converting ANNs into SNNs (Bu et al., 2021; Li et al., 2021; Ding et al., 2021; Rueckauer et al., 2017; Diehl et al., 2015). Consequently, ANNs serve as foundational pillars for contemporary SNNs. Reflecting this trend, numerous SNN libraries have been developed atop the PyTorch framework, each distinguished by its unique focus (Hazan et al., 2018; Fang et al., 2020; Pehle & Pedersen, 2021; Eshraghian et al., 2021).

While SNNs exhibit promise, a significant challenge arises with the incorporation of the temporal dimension. This addition often leads to slower training speeds. Many researchers have sought to mitigate this by truncating the network's temporal length (Chowdhury et al., 2021; Suetake et al., 2023; Xu et al., 2023), inadvertently suppressing its temporal characteristics and making SNNs closely resemble ANNs (Han et al., 2020; Bu et al., 2023), which potentially forfeits richer research opportunities. Nonetheless, the relatively sluggish execution speeds of SNNs on prevalent GPU platforms impede consistent research progress. While specialized hardware designs can potentially address this, the prevailing lack of enthusiasm in algorithmic research diminishes motivations for

---

[1]The source code will be made publicly available.

such pursuits. In this work, we seek to address this challenge by proposing a theoretically-grounded solution that parallelizes the temporal information of SNNs during propagation. The proposed approach not only mitigates the slowdown induced by the temporal dimension but also paves the way for the scalable expansion of SNNs in distributed GPU environments. In summary, our primary contributions include:

- We have proposed a temporal parallelization method tailored for universal SNN units. From the theoretical point of view, we propose to decouple the propagation pattern of the generic SNN model in the temporal dimension, streamlining the information transfer while preserving arithmetic accuracy. The method eschews the need for approximation substitutions, ensuring both simplicity and fidelity in the model's outcomes.

- We have extended the proposed parallelization method across multiple GPUs. Our multi-GPU solution diverges from conventional batch-based distribution, aligning instead with a Multiple Instruction, Multiple Data (MIMD) distributed architecture. This unique design, inherently suited to SNNs, distributes the model across GPUs temporally. Theoretically, with sufficiently large timesteps, this architecture facilitates progressive speedup benefits commensurate with timestep expansions.

- We have designed and implemented specialized acceleration infrastructures based on both CUDA and JAX frameworks. Leveraging our refined propagation pattern with reduced temporal dependencies, we have crafted implementations that amalgamate parallelism with operator fusion. The implementations optimize performance across varied timesteps by minimizing memory access overhead. Empirical evaluations substantiate the notable performance gains achieved in both CUDA and JAX environments.

## 2 BACKGROUND

As the fundamental units of the brain, neurons exhibit unique information transfer properties. To emulate these intricate behaviors in computational models, various spiking neuron models have been proposed. Notably, the Spike Response Model (SRM) Gerstner et al. (2014) offers an accurate representation for a broad category, encompassing parameters like membrane potential decay, spike threshold, refractory period, etc. Essentially, the membrane potential of a neuron undergoes continuous decay unless it receives an external stimulus. Upon receiving information, the potential increases until it hits a threshold, resulting in the generation of a spike, followed by an immediate potential drop.

However, as the deep learning domain advances, the need for simplicity becomes paramount. Complex models, while accurate, often introduce implementation challenges. Notably, from a statistical perspective, SRM can be viewed as a Generalized Linear Model (GLM) (Truccolo et al., 2005), thus enabling its simplification. The Leaky Integrate-and-Fire (LIF) model emerges as a streamlined version of SRM, preserving its core information transfer mechanisms while reducing the intricacies of neuronal information transmission. Recognizing the potential of this simplification, recent research has further refined the LIF model, yielding an iterable form (Wu et al., 2019). This adaptation ensures that spiking neurons seamlessly integrate into the deep learning paradigm without compromising the salient features of SRM.

Consider a deep spiking neural network (SNN) with multiple layers. For the $n$-th layer, let $v_i^{(t,n)}$ denote the membrane potential of the $i$-th neuron at time $t$, and let $x_i^{(t,n)}$ represent its corresponding output. We assume that, at the initial time step, the membrane potential for each neuron in layer $n$ is given by $v_i^{(0,n)} = V_{\text{rest}}^{(n)}$, where $V_{\text{rest}}^{(n)}$ characterizes the resting potential of a neuron in that layer. Additionally, let $k_\tau$ be a decay factor governing the potential's evolution over time. Under these assumptions, the iterative dynamics of the LIF model for each neuron can be mathematically formulated as:

$$v_i^{(t,n)} = k_\tau \cdot v_i^{(t-1,n)} \cdot \left(1 - x_i^{(t-1,n)}\right) + x_i^{(t-1,n)} \cdot V_{\text{rest}}^{(n)} + x_i^{(t,n-1)}, \tag{1}$$

with the spike generation mechanism as:

$$x_i^{(t,n)} = \Theta\left(v_i^{(t,n)} - V_{\text{th}}^{(n)}\right) = \begin{cases} 1, & \text{if } v_i^{(t,n)} - V_{\text{th}}^{(n)} \geq 0 \\ 0, & \text{otherwise} \end{cases}. \tag{2}$$

Here, $\Theta$ signifies the spike activation function intrinsic to SNNs, with $x_i^{(t,n-1)}$ being the output from the neuron in the preceding layer. Since these formulations provide a rigorous representation of the spiking mechanism and preserve the inherent characteristics of spiking neurons, without loss of generality, the ensuing discussions and methodologies in this paper are grounded on them.

## 3 PARALLELIZATION METHOD

We first delve into the foundational aspects of the the tailored parallelization method and its theoretical implications when executed on a single GPU. Then, it is further followed by an exploration of the method's scalability across multiple GPUs.

### 3.1 TEMPORAL PARALLELIZATION ON SINGLE GPU

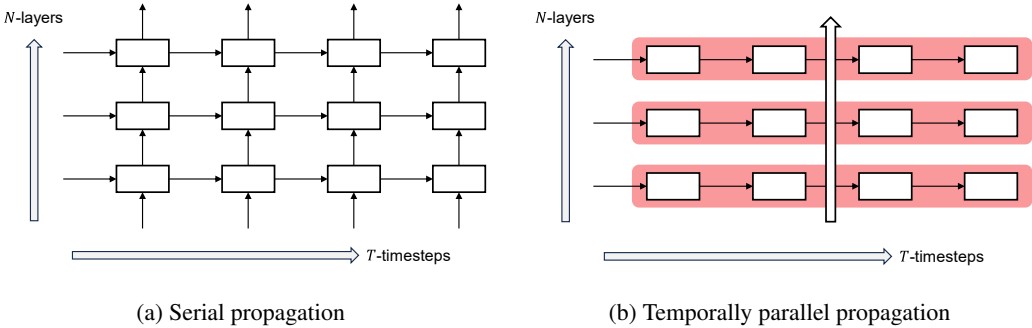

(a) Serial propagation        (b) Temporally parallel propagation

Figure 1: Comparative illustration of the conventional serial propagation model versus the temporally parallel propagation approach for SNNs. The temporal dimension is unfolded, and each individual box symbolizes the collection of neuronal data for a specific SNN layer at a given time instance.

As given by Eq. (1), the dynamics of the membrane potential in SNNs are intricately tied to both temporal and spatial factors. This entails that the activity of any given neuron at time $t$ is predicated on its preceding state, as well as the input it receives from neighboring neurons within the same layer. Fig. 1a graphically demonstrates this spatial-temporal interplay by unfolding the temporal dimension. In this representation, each encapsulated set of neuronal information in a box is contingent on its antecedent data in both the time and spatial dimensions.

A noteworthy observation is that the computations within the same network layer across various time moments are fundamentally equivalent. This inherent characteristic offers an avenue for parallelization across the temporal dimension, provided that the initial states for all time steps are pre-established. To actualize this parallelism, a synthesis of Eq. (1) and Eq. (2) yields the following formulation:

$$v_i^{(t,n)} = \underbrace{k_\tau \cdot v_i^{(t-1,n)} + \left(V_{\text{rest}}^{(n)} - k_\tau \cdot v_i^{(t-1,n)}\right) \cdot \Theta\left(v_i^{(t-1,n)} - V_{\text{th}}^{(n)}\right)}_{f^{(n)}\left(v_i^{(t-1,n)}\right)} + x_i^{(t,n)}. \tag{3}$$

As visualized in Fig. 2, this derived equation decouples the dependency of $v_i^{(t,n)}$ at time $t$ from $x_i^{(t-1,n)}$ at the preceding moment $t-1$. Such a transformation is instrumental in facilitating parallel computation.

From Eq. (3), in any layer $n$ at time $t$, the membrane potential $v_i^{(t,n)}$ is effectively described by $f\left(v_i^{(t-1,n)}\right)$ and the associated output $x_i^{(t,n-1)}$. If all inputs across various time points are immediately available, then $x_i^{(t,n-1)}$ can be computed in a parallel manner, thereby optimizing the computational effort.

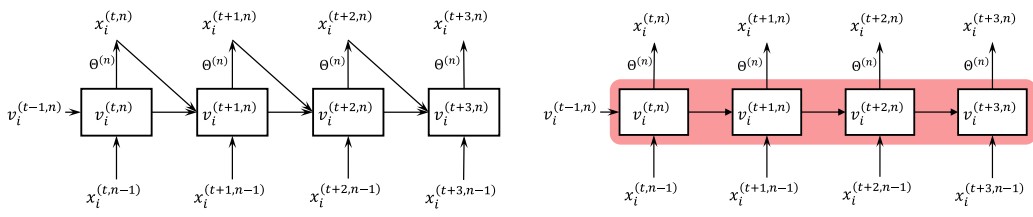

(a) Data flow of serial propagation        (b) Data flow of temporally parallel propagation

Figure 2: Transformation from the conventional serial propagation methodology to a temporally parallelized approach in SNNs. The left panel showcases the data flow based on the original SNNs propagation model as per Eq. (1), while the right panel depicts the enhanced flow after the transformation as detailed in Eq. (3). This transformation retains the integrity of the final computation.

It is essential to note that each $v_i^{(t,n)}$ arises from a nonlinear evaluation of $f(\cdot)$ based on its prior time value $v_i^{(t-1,n)}$, making it non-trivial to decouple the data flow of $v_i$. This challenge, however, can be addressed using operator fusion techniques. T his approach is visualized in Fig. 1b and the right segment of Fig. 2, where the rounded boxes consolidate computations for a layer over time, thus reducing repetitive memory operations.

Theoretically, for a specific layer in the SNNs having a timestep of $T$ and given that the average duration for accessing memory of a single timestep's data is $t_{\mathrm{mem1}}$, conventional SNN propagation would necessitate the GPU to undertake reading and writing operations for $T$ data sets. In contrast, with the benefits of parallel optimization and operator fusion, the GPU can simultaneously read and write $T$ data sets, thus resulting in only two memory operations. If the average time taken to simultaneously access $T$ data sets is $t_{\mathrm{mem2}}$, the total durations for serial and parallel access, denoted by $t_{\mathrm{s}}$ and $t_{\mathrm{p}}$ respectively, become:

$$t_{\mathrm{s}} = 2 \cdot T \cdot t_{\mathrm{mem1}}, \tag{4a}$$

$$t_{\mathrm{p}} = 2 \cdot t_{\mathrm{mem2}}. \tag{4b}$$

In an idealistic scenario where other overheads are negligible, the maximum potential speedup ratio, denoted as $\sigma$, can be expressed as:

$$\sigma = \frac{t_{\mathrm{s}}}{t_{\mathrm{p}}} = \frac{T \cdot t_{\mathrm{mem1}}}{t_{\mathrm{mem2}}}. \tag{5}$$

## 3.2    PARALLELIZATION ACROSS MULTIPLE GPUS

The method introduced in the previous subsection facilitates temporal parallelism in SNNs. By reducing memory access and utilizing operator fusion, this method achieves efficient computation, even with small timesteps. However, when the timestep becomes significantly large, computation on a single GPU can become a bottleneck. As the proposed approach expands the temporal dimension, it may lead to storage limitations for large timesteps. Consequently, if only one GPU is at hand, the system may revert to a more conventional serial propagation pattern. To mitigate this limitation, this section explores the feasibility of extending the temporal parallelization method across multiple GPUs.

Let us assume there are $k$ $(k \geq 2)$ GPUs available. If the computational workload of the SNNs for one layer is extensive enough to be split into $k$ non-parallelizable segments in terms of the time dimension, then the time required to complete $k$ sub-tasks on these GPUs is $t^{(k)}$. If the inter-GPU communication time is $T_{\mathrm{c}}$, then we have:

$$T_{\mathrm{s}} = k \cdot t^{(k)}, \tag{6a}$$

$$T_{\mathrm{m}}^{(k)} = (k-1) \cdot T_{\mathrm{c}} + t^{(k)}, \tag{6b}$$

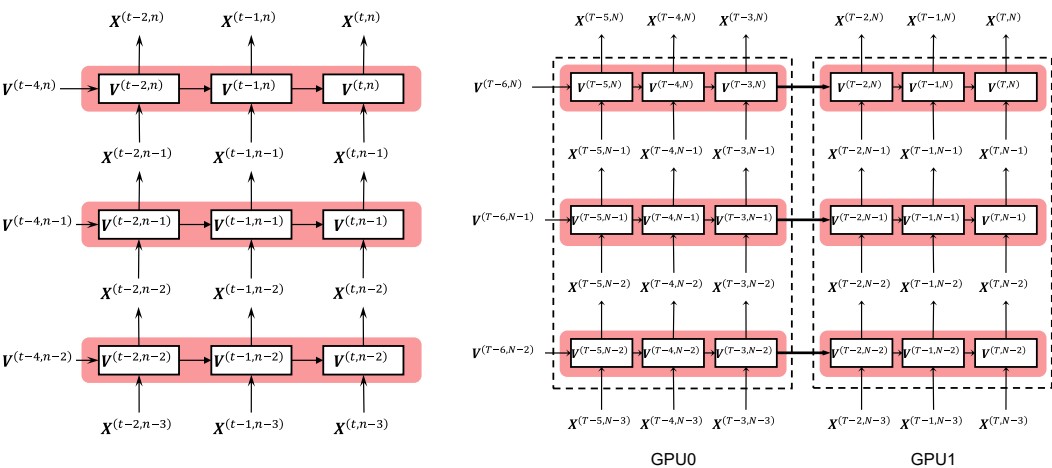

(a) Temporal parallelization on a single GPU      (b) Temporal parallelization across multiple GPUs

Figure 3: Schematics showcasing temporal parallelization on both single and multiple GPUs. The horizontal axis represents time, while the vertical axis represents the hierarchical propagation of the networks. The capitalized forms $\mathbf{V}^{(t,n)}$ and $\mathbf{X}^{(t,n)}$ represent the membrane potential and neuronal output in the tensor form of SNNs, corresponding to the element-wise variables $v_i^{(t,n)}$ and $x_i^{(t,n)}$ in Eq. (1).

where $T_{\mathrm{s}}$ and $T_{\mathrm{m}}^{(k)}$ denote the total task time on a single GPU and $k$ GPUs respectively. Correspondingly, the multi-GPU speedup rate $\mu$ can be defined as:

$$\mu = \frac{T_{\mathrm{s}}}{T_{\mathrm{m}}^{(k)}} = \frac{k}{(k-1) \cdot T_{\mathrm{c}} / t^{(k)} + 1}. \tag{7}$$

As Eq. (7) indicates, the speedup ratio $\mu$ will exceed 1 whenever $t^{(k)}$ surpasses $T$. This suggests that leveraging multi-GPUs will yield a performance boost, especially when tasks that are non-parallelizable on a single GPU are time-intensive.

For SNNs consisting of a single layer of spiking-activated neurons, there is only one communication instance between two GPUs. This communication can be approximated as a constant $T_{\mathrm{c}}$ as shown in Eq. (6b). However, in more complex scenarios with multiple spiking neuron layers (as depicted in Fig. 3b), the data transmission time between layers might vary due to different layer data sizes. This variability introduces opportunities for further optimization in inter-GPU transfers. For multi-layered spiking neurons, asynchronous multi-GPU communication can be employed to ensure $T_{\mathrm{c}} \in (T_{\mathrm{lb}}, T_{\mathrm{ub}}]$, with:

$$T_{\mathrm{lb}} = T_{\mathrm{cmax}}, \tag{8a}$$

$$T_{\mathrm{ub}} = N \cdot T_{\mathrm{avg}}, \tag{8b}$$

where $N$ denotes the number of layers of spiking neurons, $T_{\mathrm{cmax}}$ and $T_{\mathrm{avg}}$ denote the maximum value of inter-GPU transmission elapsed time and the transmission elapsed time expectation in neurons of $N$ layers, respectively.

## 4   PROGRAMMING MODELS AND IMPLEMENTATIONS

This section introduces the infrastructures developed around the temporal parallelization method. Specifically, the implementations leverage the foundational architectures of JAX(Bradbury et al., 2018) and CUDA (NVIDIA et al., 2023). While JAX implementation supports single GPU parallelization, the CUDA implementation accommodates both single and multi-GPU configurations. We begin by discussing the programming models tailored for both JAX and CUDA implementations, followed by details about the implementation schemes and an exploration of the design considerations for the associated functional modules.

## 4.1 Programming Models

This subsection presents the programming models for both JAX (single-GPU) and CUDA (single and multi-GPU). Examples will be used to illustrate the interfaces provided by our infrastructures, named `snn_jax` and `SnnCuda` for JAX and CUDA, respectively.

```python
from jax import random
from flax.linen import Conv
from snn_jax import LIF, snn_wrapper

seed        = ... # User-defined random number seed
input_shape = ... # User-defined input data size
datasets    = ... # User-defined datasets
conv_info   = ... # User-defined convolutional layer information
lif_info    = ... # User-defined LIF layer information
rng, subkey = random.PRNGKey(seed)

# User-defined spiking neural network
network = snn_wrapper((
    fnn.Conv(conv_info),
    LIF(lif_info)
), subkey, input_shape)

# Use of user-defined spiking neural network
for data in datasets:
    logits = network(data)
```

Listing 1: Python example of the single-GPU programming model using JAX with Flax. `snn_jax` is the custom infrastructure package.

Listing 1 provides an example based on the JAX programming model. Rooted in Python, the example demonstrates how a convolutional layer combined with a LIF neuron can be used to design SNNs optimized for single GPU temporal parallelization. The `snn_wrapper` encapsulates the network layers, making the neural network usable as a straightforward function call.

Listing 2 presents the CUDA-based programming model, grounded in C#. This object-oriented approach offers encapsulated interfaces. Much like the JAX model, the user defines specific network parameters before running it. The example showcases a convolutional layer paired with a LIF neuron, with both single and multi-GPU temporal parallelization abstracted from the user through the `SnnCuda` namespace.

## 4.2 Implementations

In the proposed infrastructure, the primary focus is on optimizing spiking neurons at the design and implementation levels. However, modern SNNs usually consist of both ANN operators (e.g. convolution) and spiking neurons. Hence, to construct complete SNNs, we integrated existing frameworks. For the JAX implementation, we employed the ANN operators from Flax (Heek et al., 2023). For the CUDA implementation, we incorporated the ANN operators directly from the cuDNN framework (Chetlur et al., 2014).

For operator fusion in the JAX implementation, we utilized the just-in-time (JIT) module. Specifically, we incorporated the computational-graph-level compiler, XLA, which offers substantial operator fusion capabilities. On the other hand, for the CUDA implementation (both single and multi-GPU configurations), we employed basic native functions provided by both C# and CUDA. This pre-compilation approach enhances the speedup, thus yielding further performance gains.

```
1   using SnnCuda;
2
3   class Program {
4     static void Main(string[] args) {
5       long seed = ...; // User-defined random number seed
6
7       /* User-defined datasets and input data size */
8       GPUOffsetPointer<T> datasets =  ...;
9       readonly var inputShape     => ...;
10
11      /* User-defined layer information */
12      readonly var convInfo => ...;
13      readonly var lifInfo  => ...;
14
15      /* User-defined spiking neural network */
16      var network = new List<ILayer> {
17        ConvLayer(convInfo),
18        LIFLayer(lifInfo)
19      };
20
21      /* Single-GPU computing */
22      int device = ...;
23      var info = new NetInfo(network, inputShape, seed, device);
24      foreach (var data in datasets) {
25        GPUScheduler.TParallelSingle<T>(info, data, out logits);
26      }
27
28      /* Multi-GPU computing */
29      List<int> devices = ...;
30      var info = new NetInfo(network, inputShape, seed, devices);
31      foreach (var data in datasets) {
32        GPUScheduler.TParallelMultiple<T>(info, data, out logits);
33      }
34    }
35  }
```

Listing 2: C# example of the single and multi-GPU programming model using CUDA. `SnnCuda` is the custom infrastructure package.

## 5 EXPERIMENTS

### 5.1 EXPERIMENTAL SETTINGS

The experiments in this section involve runtime comparisons across multiple platforms and implementations. To ensure fair comparisons, we standardized the hyperparameters as per Eq. (1), setting $V_{rest} = 0$, $k_\tau = 0.2$, and $V_{th} = 0.3$. For libraries which do not provide hyperparameters that include $k_\tau$, we use $\tau = 1.25$ with $k_\tau = 1 - 1/\tau$. All experiments were conducted on unoccupied A100 GPUs to ensure consistent performance metrics.

### 5.2 SINGLE-GPU ACCELERATION

In this part, we conduct experiment on single-GPU acceleration. Specifically, the experiment is conducted on a single-layer LIF neuron with each layer containing 1,000,000 neurons. The the final results are averaged over 1,000 independent executions.

As evidenced in Table 1, our approach yields promising acceleration performance on both JAX and CUDA implementations. Notably, the CUDA implementation demonstrate a more significant speedup, achieving approximately 40x speedup than other SNN implementations.

| Implementation | Timesteps | | | | |
|---|---|---|---|---|---|
| | **8** | **16** | **32** | **64** | **128** |
| PyTorch | 0.6460 | 1.6525 | 3.0774 | 4.4984 | 9.6468 |
| SpikingJelly-PyTorch | 0.4987 | 0.8368 | 1.4091 | 2.5561 | 4.8794 |
| SpikingJelly-CuPy | 0.4923 | 0.8123 | 1.4094 | 2.5398 | 4.8246 |
| snnTorch | 2.0784 | 4.0026 | 7.6813 | 15.1730 | 30.5006 |
| JAX (**ours**) | **0.1130** | **0.1749** | **0.3174** | **0.6043** | **1.1839** |
| CUDA (**ours**) | **0.0542** | **0.1005** | **0.1934** | **0.3813** | **0.7631** |

Table 1: Single-GPU performance of our approach against other SNN implementations. Here, Py-Torch represents a baseline serial approach. All time measurements are in seconds.

## 5.3 MULTI-GPU ACCELERATION

Figure 4: Performance of multi-GPU acceleration using the CUDA implementation of our approach.

In this part, we conduct experiment on multi-GPU acceleration. Specifically, the experiment is conducted with the number of GPUs ranging from 1 to 4. The the final results are averaged over 1,000 independent executions. As shown in Fig. 4, the computational costs reduce significantly as the number of GPUs increase, indicating the promising performance of our approach across multiple GPUs.

## 6 RELATED WORK

### 6.1 SNN LIBRARIES

In light of the integration of spiking neuron properties into deep learning, a plethora of libraries tailored for SNNs have emerged, each accentuating different facets of the domain. BindsNET (Hazan et al., 2018), built upon the PyTorch platform, emphasizes applications in machine and reinforcement learning. SpikingJelly (Fang et al., 2020), on the other hand, accentuates algorithmic advancements in SNNs, offering two backends and two SNN propagation mechanisms, thereby fostering both algorithmic enhancements and performance optimizations. Norse (Pehle & Pedersen, 2021) and snnTorch (Eshraghian et al., 2021) expand on SNNs within the PyTorch ecosystem, prioritizing comprehensive functional support alongside an extensive documentation suite, lauded for its user-centric design. In contrast, SPAIC (Hong et al., 2022) and ENLARGE (Qu et al., 2023) take a distinctive approach, leaning towards a more detailed portrayal of neural properties and offering an alternative programming paradigm. Furthermore, libraries like GeNN (Yavuz et al., 2016) and cuSNN (Paredes-Valles et al., 2020), developed directly atop CUDA, minimize runtime overhead by leveraging substrate-based designs.

### 6.2 GPU-ACCELERATION FRAMEWORKS

In the GPU acceleration landscape, NVIDIA's CUDA (NVIDIA et al., 2023) stands out by offering a high-level interface for direct GPU interactions. As a foundational software layer, CUDA facilitates direct engagement with the GPU's virtual instruction set and its parallel computing components, thereby streamlining kernel computations. This closeness to the hardware layer augments development flexibility. To further simplify the developer experience, optimized standard routines are provided. For instance, cuDNN (Chetlur et al., 2014) delivers deep neural network operator implementations within the CUDA framework. Many high-level deep learning frameworks are built on this foundation.

By harnessing the power of Autograd and XLA, Google's JAX (Bradbury et al., 2018) furnishes an interface tailored for high-performance numerical computations and deep learning. Its just-in-time (JIT) compilation mechanism translates profiled computational graphs into optimized machine code, enhancing both execution efficiency and performance. In the realm of deep neural network

training, JAX offers notable performance uplifts. The Flax deep learning framework, inheriting JAX's capabilities, leverages its Autograd and JIT features to enhance the performance of deep learning models.

## 7 CONCLUSION

In this work, we introduced a versatile GPU-accelerated technique for the temporal parallelization of SNNs, demonstrating its efficacy on both single and multi-GPU architectures. Rooted in the core principles of neuroscience, we specifically targeted spiking neurons compatible with deep learning paradigms, ensuring the preservation of intrinsic properties in line with existing methodologies. This foundational understanding enabled the design of parallelization and operator fusion optimization schemes tailored to the unique propagation characteristics of SNNs. Our approach was then naturally extended to encompass multi-GPU architectures, underpinned by a thorough theoretical analysis. We subsequently implemented single-GPU infrastructures built on both JAX and CUDA platforms, as well as a multi-GPU solution leveraging the CUDA framework, each accompanied by intuitive programming models. Benchmarked against some prominent implementations of SNNs, the comprehensive experiments underscored the versatility and efficiency gains of our proposed infrastructures.

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
