# OpenReview forum: "Temporal Parallelization for GPU Acceleration of Spiking Neural Networks"
_ICLR.cc/2024/Conference — ICLR 2024 Conference Withdrawn Submission_

### Official Review · Reviewer_yZKs · 2023-10-27

**Soundness:** 3 good
**Presentation:** 1 poor
**Contribution:** 2 fair
**Rating:** 3
**Confidence:** 5

**Summary:**

This paper proposes a temporal parallelization method for SNNs that can accelerate SNNs on both single or multi GPUs with up to 40x acceleration.

**Strengths:**

The training of deep SNNs requires much more time and memory consumption. Thus, it is meaningful to explore the acceleration of simulating SNNs on GPUs.

**Weaknesses:**

The details of the proposed method are not described clearly in this paper. To make matters worse, the Supplementary Material is the same as the main paper.

**Questions:**

In Figure 3, how the propagation of the spiking neuron layer is paralleled? I assume that V[t] is still computed in serial. For an input sequence with length T, the time complexity is still O(T).

In section 3.2, the authors claim that the SNN accelerated by pipeline in multiple GPUs may have a faster speed than using a single GPU. However, I am afraid that the communication time between GPUs will be the bottleneck. According to my experience, the communication time is much longer than any other time. Thus, the pipeline method is seldom used in training, and the Distributed Data Parallel is the mainstream.

In Table 1, the time of SpikingJelly with or without CuPy does not have much difference, which is against my experience.


Where is Figure 4?

---

### Official Review · Reviewer_VQqA · 2023-10-30

**Soundness:** 3 good
**Presentation:** 2 fair
**Contribution:** 2 fair
**Rating:** 3
**Confidence:** 4

**Summary:**

This paper presents an SNN-based acceleration strategy with parallelized temporal computation that supports both single and multiple GPUs.

**Strengths:**

Largely improved SNN inference speed compared to the previous implementation.

Compatibility with both single and multi-GPU processing.

**Weaknesses:**

**W1:** Figure 4 is missing.

**W2:** The biggest bottleneck of this work is that the accuracy benchmarking is completely missing in the paper. I understand the inference speed-up is very important in SNN, but I cannot see the reason why the paper chose not to report the accuracy. It is important to verify the proposed implementation with different SNN model architectures. E.g.. ResNet vs. VGG.

**W3:** It seems like the implementation can only accelerate the inference rather than training, which I think is not powerful enough.

**Questions:**

Please refer to the Weakness.

---

### Official Review · Reviewer_4kmC · 2023-10-31

**Soundness:** 2 fair
**Presentation:** 1 poor
**Contribution:** 2 fair
**Rating:** 3
**Confidence:** 4

**Summary:**

This paper trys to use a temporal parallelization method to accelerate the propagation dynamics of SNNs on GPUs. The feature it claims is a cross-timestamp acceleration of LIF model. With the Leaky Integrate- and-Fire model as a test case, the CUDA-based implementation achieved 5× to 40× acceleration on the A100 GPU.

**Strengths:**

The author proposed temporal parallelisation method tailored for universal SNN units on single and multiple GPUs. It supports both CUDA and JAX frameworks.

**Weaknesses:**

1. The motivation behind this paper lacks clarity. Spiking Neural Networks (SNNs) are not typically intended for deployment on GPUs, meaning that a GPU is not the most suitable platform for SNN deployment. Without a demonstration of the clear benefits of utilizing GPUs for SNNs deployment as opposed to other platforms, the paper's underlying motivation remains unconvincing.

2. How does this paper leverage GPU to implement true spiking mechanism? It is not clear or discussed. Is it only considering simulating the mechanism of the Leaky-Integrate-and-Fire model behavior? Plus, there’s no true spiking signals in GPU, addressing the temporal information is not really Spiking implementation. This paper doesn’t clarify the basic concept.
(In some sense, parallelizing temporal information is possible, but there’s conversion between spiking temporal information and the muti-bit digital temporal information for GPU? Then what’s the conversion cost?)

3. Although this paper is based on the computational model of LIF, but it does not clearly describe how training and inference is done, respectively. Training an SNN is hard, and it is not discussed at all in this paper, so it’s only about inference, or even, the simulation of inference?

4. Last but not least, most importantly, this paper does not provide any AI-model based results, such as accuracy, performance, respective speed-up, etc, let alone thorough analysis based on the comparison of results. The only result is a table based on a single-layer toy model? For multi-GPU, where is Fig. 4, seems this paper is incomplete?

**Questions:**

1. How SNN neuron spiking behaviour described in Eq.1 and Eq. 2 reflected in GPU?

---

### Official Review · Reviewer_81ci · 2023-10-31

**Soundness:** 1 poor
**Presentation:** 2 fair
**Contribution:** 2 fair
**Rating:** 3
**Confidence:** 3

**Summary:**

The authors describe a method and code for accelerating spiking neural
networks (SNNs) on GPUs. They first claim to parallelize the temporal
membrane integration of a layer in an SNN and secondly divide the
compute onto multiple GPUs. They provide a template how to implement
it in common ML frameworks such as JAX. Finally, they show that their
implementation outperforms other toolboxes.

**Strengths:**

The implementation seems to outperform current toolboxes in terms of runtime.  The authors show that this layer-first
approach gives a considerable speedup on GPUs due to reduced memory movement.

**Weaknesses:**

While better implementation of simulating SNNs using GPUs has its merits,
the task is merely a software
engineering task. The paper does not add any value in terms of novel
insights. This is in particular true since the "temporal
parallization" argumentation is indeed a misnomer as the temporal dimension is *not*
computed in parallel, but instead time of one layer is simply handled
within one GPU kernel (but still computed sequentially if I understand it correctly).

If one wanted to design a custom CUDA kernel and would assume that
only feed-forward layers are allowed, this would be just the standard
approach to do, I don't see any innovative aspects here. In
particular, equation 3 is just a re-writing (inserting) of $x^{(t-1, n)}$,
there is no "transformation" I can see. Note that $v_i^{(t, n)}$ still
is a function of previous times, $v_i^{(t-1, n)}$. All what is done is to
compute all time steps per layer first before sending the full output
spike train to the next layer. This will obviously not work for
recurrent SNNs.

Also, the authors do not even provide their own optimized CUDA kernel
(which would have more merit), but instead rely on generic toolboxes
like JAX. The code listings do not provide any details of the
implementation and are more like a tutorial how to use it.

Overall, while the implementation might be useful as it improves the
runtime of SNNs compared to the (apparently very non-optimized)
standard SNN packages, the paper does not provide any new scientific
insights. It is also not discussed that the approach works only for
feed-forward SNNs. Moreover, the presentation of "temporal
parallelization" is not correct (as it just points to a fused
sequential CUDA-kernel). Finally, the layer-first approach (fusing
kernels to reducing memory operations) and dividing the compute for
multiple GPUs are rather standard practices in GPU programming in
general and not novel enough for a research oriented conference
contribution in my opinion.

**Questions:**

*  In Eq 3: $x_i^{(t, n)}$ should be $x_i^{(t, n-1)}$